# COVID-19 Associated Pulmonary Aspergillosis (CAPA): Hospital or Home Environment as a Source of Life-Threatening *Aspergillus fumigatus* Infection?

**DOI:** 10.3390/jof8030316

**Published:** 2022-03-19

**Authors:** Teresa Peláez-García de la Rasilla, Irene González-Jiménez, Andrea Fernández-Arroyo, Alejandra Roldán, Jose Luis Carretero-Ares, Marta García-Clemente, Mauricio Telenti-Asensio, Emilio García-Prieto, Mar Martínez-Suarez, Fernando Vázquez-Valdés, Santiago Melón-García, Luis Caminal-Montero, Inmaculada Fernández-Simón, Emilia Mellado, María Luisa Sánchez-Núñez

**Affiliations:** 1Microbiology Department, Central University Hospital of Asturias (HUCA), 33011 Oviedo, Spain; opsklins@gmail.com (F.V.-V.); santiago.melon@sespa.es (S.M.-G.); 2Biosanitary Foundation for Research in the Principality of Asturias (FINBA), 33011 Oviedo, Spain; 3Mycology Reference Laboratory, National Center for Microbiology, Carlos III Health Institute (ISCIII), 28220 Madrid, Spain; irene.gojim@gmail.com (I.G.-J.); alex7799roldan@gmail.com (A.R.); emellado@isciii.es (E.M.); 4Preventive Medicine Department, Central University Hospital of Asturias (HUCA), 33011 Oviedo, Spain; andreagfarroyo@gmail.com (A.F.-A.); carreteroj@outlook.es (J.L.C.-A.); mar.martinezsuarez@gmail.com (M.M.-S.); 5Pneumology Department, Central University Hospital of Asturias (HUCA), 33011 Oviedo, Spain; 6Infectious Diseases Department, Central University Hospital of Asturias (HUCA), 33011 Oviedo, Spain; mauritelenti@gmail.com (M.T.-A.); lcaminal@gmail.com (L.C.-M.); 7Intensive Care Unit Department, Central University Hospital of Asturias (HUCA), 33011 Oviedo, Spain; egarciaprieto@gmail.com (E.G.-P.); ifernandezsimon@gmail.com (I.F.-S.); 8Spanish Network for Research and Infectious Diseases (REIPI RD16/CIII/004/0003), 28220 Madrid, Spain; 9Hospital Direction, HUCA, 33011 Oviedo, Spain; marisasanchezn@sespa.es

**Keywords:** COVID-19, aspergillosis, CAPA community-acquired, CAPA hospital-acquired

## Abstract

Most cases of invasive aspergillosis are caused by *Aspergillus fumigatus*, whose conidia are ubiquitous in the environment. Additionally, in indoor environments, such as houses or hospitals, conidia are frequently detected too. Hospital-acquired aspergillosis is usually associated with airborne fungal contamination of the hospital air, especially after building construction events. *A. fumigatus* strain typing can fulfill many needs both in clinical settings and otherwise. The high incidence of aspergillosis in COVID patients from our hospital, made us wonder if they were hospital-acquired aspergillosis. The purpose of this study was to evaluate whether the hospital environment was the source of aspergillosis infection in CAPA patients, admitted to the Hospital Universitario Central de Asturias, during the first and second wave of the COVID-19 pandemic, or whether it was community-acquired aspergillosis before admission. During 2020, sixty-nine *A. fumigatus* strains were collected for this study: 59 were clinical isolates from 28 COVID-19 patients, and 10 strains were environmentally isolated from seven hospital rooms and intensive care units. A diagnosis of pulmonary aspergillosis was based on the ECCM/ISHAM criteria. Strains were genotyped by PCR amplification and sequencing of a panel of four hypervariable tandem repeats within exons of surface protein coding genes (TRESPERG). A total of seven genotypes among the 10 environmental strains and 28 genotypes among the 59 clinical strains were identified. Genotyping revealed that only one environmental *A. fumigatus* from UCI 5 (box 54) isolated in October (30 October 2020) and one *A. fumigatus* isolated from a COVID-19 patient admitted in Pneumology (Room 532-B) in November (24 November 2020) had the same genotype, but there was a significant difference in time and location. There was also no relationship in time and location between similar *A. fumigatus* genotypes of patients. The global *A. fumigatus,* environmental and clinical isolates, showed a wide diversity of genotypes. To our knowledge, this is the first study monitoring and genotyping *A. fumigatus* isolates obtained from hospital air and COVID-19 patients, admitted with aspergillosis, during one year. Our work shows that patients do not acquire *A. fumigatus* in the hospital. This proves that COVID-associated aspergillosis in our hospital is not a nosocomial infection, but supports the hypothesis of “community aspergillosis” acquisition outside the hospital, having the home environment (pandemic period at home) as the main suspected focus of infection.

## 1. Introduction

Exposure to some fungal species, such as *Aspergillus fumigatus*, has been associated with opportunistic invasive fungal infections, a significant cause of mortality and morbidity in patients with severe neutropenia or immunosuppression. In these immunosuppressed patients, invasive aspergillosis (IA) is easy to suspect because given the typical and well-described risk factors and forms of presentation [1]. In recent years, several case series have been published about patients with severe influenza virus disease and without other risk factors who develop IA. These are generally patients with severe acute respiratory distress syndrome, in which incidence of IA depends on geographical differences, vaccination rates, and diagnostic tests used, varying from 11 to 28% [2,3,4]. More recently, in relation to the COVID-19 pandemic, an increasing number of patients with SARS-CoV-2 pneumonia and respiratory distress have been diagnosed with IA [2]. According to the different studies published, CAPA prevalence is highly variable, ranging between 3 and 33% [5,6,7,8,9]. These differences can be attributed to different thresholds for clinical evaluation, diagnostic methods, and differences in criteria for CAPA definition [6,10]. Prattes et al. [5], in a multicenter study, described a median prevalence of 10.7%. Our group has published a recent study with a prevalence of 11.7% during the first and second waves of pandemic COVID-19 [11]. This entity has been reported to significantly increase the severity of the COVID-19 with worse outcome and it has an important impact on the prognosis of the SARS-CoV-2 infection [2,6,7,8,9], so an early diagnosis can influence survival.

The growing number of cases has led to the belief in the possibility of nosocomial transmission of IA. Nosocomial acquisition of invasive aspergillosis has been proved in epidemic situations such as construction and renovation works, and also in non-epidemic cases [1,4,12,13,14], with studies showing how environment concentrations of *Aspergillus* and other fungi can be correlated with the presence of IA [1,14]. For this reason, during hospitalization, severely immunocompromised patients are believed to be at high risk of IA and, therefore, should receive care in units with rooms equipped with high-efficiency filters, laminar flow, and positive pressure [15] while maintaining good air quality in hospitals also contributes significantly to reducing the incidence of these *Aspergillus* infections [16]. However, the presence of filamentous fungi in the external environment has also been reported to have a significant impact on the disease [17,18], and therefore the influence of external isolates, not related to hospital nosocomial infections, as a source of IA has raised interest.

Our hospital saw a rapid increase of IA in patients with COVID-19 (CAPA) during the first and second waves of the pandemic, which led to an active search for cases and an interest to discover whether there was an element of nosocomial transmission to these infections [11]. Different publications have reported different incidences of CAPA among hospitalized patients [5,6,7,8,9], and none of them have been able to differentiate nosocomial and community infections, which is important for case investigation and infection control since hospital-acquired infections require adequate control measures to prevent subsequent cases. However, it is not easy to find an association between a hospital stay and the appearance of IA since it depends on the timing and the biological possibility. Therefore, epidemiological criteria must be assessed, as well as the time between admission and the onset of symptoms and microbiological criteria.

It is also important to bear in mind that while it is already difficult to establish a nosocomial origin in the case of common bacterial pathogens, it is even more complicated in the case of *Aspergillus*, in which patients have underlying conditions that lead to severe immunosuppression that puts them at higher risk of both community and hospital-acquired infections [19]. It should also be noted that IA incubation is influenced by individual differences and environmental determinants, including the severity of immunosuppression and air quality. Knowing better the early events related to IA will help prevent this disease, for which prognosis continues to be poor.

The purpose of our study was to evaluate the source of infection in CAPA patients admitted to Hospital Universitario Central de Asturias during the first and second wave of the COVID-19 pandemic, analyzing the distribution of *Aspergillus* spp. species among the samples of patients diagnosed with CAPA and hospital environmental isolates and carrying out the genotyping of the *A. fumigatus* isolates, to assess whether the aspergillosis was hospital or community-acquired.

## 2. Materials and Methods

### 2.1. Study Design

A prospective study was conducted at a tertiary university hospital in Asturias (Spain) during the COVID-19 pandemic between 1 January and 31 December 2020. Our institution is a large teaching hospital attending a population of 1,000,000 inhabitants. All the 3 intensive care units have positive pressure and HEPA filters. Patients in the ICUs are followed up by a team of specialized intensivists, anesthetists, pneumologists, and cardiovascular surgeons. Microbiologists, and infectious disease and preventive specialists perform daily rounds to collaborate in diagnosing and treating infectious complications and implementing preventive measures (HUCAPA Group).

The study was conducted among 300 patients tested for COVID-19 infection who were admitted to the hospital at ICU during the first and second waves (March–May 2020 and October–December 2020).

A diagnosis of COVID-19-associated pulmonary aspergillosis (CAPA) was defined based on the 2020 European Confederation of Medical Mycology/International Society for Human and Animal Mycology (ECCM/ISHAM) [20] consensus criteria, and according to these criteria, patients were classified as proven CAPA, probable CAPA, possible CAPA or no evidence of CAPA. CAPA diagnosis was defined as the earliest date when a diagnostic feature was identified.

The study was in accordance with the Helsinki declaration and national ethical standards. The hospital research ethics committee approved the study protocol.

### 2.2. Microbiology Data Collection

Starting in January 2020, institutional recommendations were to screen patients in ICUs for fungal infections by means of:

1. Fungal cultures from respiratory samples on Sabouraud dextrose agar plates (BioMerieux, Mercy, L’Etoile, France). Fungus identification was performed by a matrix-assisted laser desorption ionization-time of flight (MALDI-TOF) mass spectrometry instrument (Bruker, Madrid, Spain), following the manufacturer’s instructions.

2. The two commercially manufactured lateral flow devices (LFDs) (AspLFD, OLM Diagnostics, Newcastle upon Tyne, UK) and Lateral Flow Assays (LFAs) (IMMY sona *Aspergillus* Galactomannan Lateral Flow Assay, IMMY, Norman, OK, USA) tests, with a visual reader that provides a semiquantitative reading and removes subjectivity when interpreting results.

3. Quantitative real-time PCR for *Aspergillus* genus as follows: DNA extraction for PCR analysis was performed on an ELITe InGenius automated platform as well as RT-PCR using the *Aspergillus* spp. ELITe MGB kit (Elitegroup, Palex, Barcelona, Spain). The DNA was extracted from a 1-mL volume of BAL fluid and was eluted in a 200-µL saline solution before DNA amplification in the same platform. RT-PCR for the *Aspergillus* genus was performed by an *Aspergillus* spp. ELITe MGB kit, which was CE-in vitro diagnostic (CE-IVD) validated on a diverse range of sample types. The target region was the ribosomal DNA18S (rDNA18S), and the human B-globin gene was used as an internal standard. The fungal DNA copy number was expressed as copies/mL in relation to a rDNA18s standard curve.

To elucidate whether CAPA patients acquired aspergillosis in the community or the hospital, we looked for the presence of *Aspergillus* in the first respiratory sample obtained from the COVID patient, even if they were not yet in the ICU. In several available patients, we even perform *Aspergillus* PCR on the same sample used for the diagnosis of COVID.

4. GM testing was performed using Platelia™ *Aspergillus* (Bio-Rad Laboratories, Madrid, Spain) with a cut-off value of ≥0.5 in serum and ≥1.0 in BAL or ≥4 in tracheal and bronchial aspirates.

5. Detection of 1,3-β-d-glucan (βDG) in serum was performed with the Wako β-glucan test (Fujifilm Wako Pure Chemical Corporation, Vircell Microbiologists, Granada, Spain). A cut-off value of 7 pg/mL was used.

### 2.3. Antifungal Drugs Susceptibility Testing

Antifungal susceptibility testing (AFST) was performed following the European Committee on Antimicrobial Susceptibility Testing (EUCAST) broth microdilution reference method 9.3.1 [21]. Antifungals used were amphotericin B (Sigma-Aldrich Química, Madrid, Spain) and the azoles itraconazole (Janssen Pharmaceutica, Madrid, Spain), voriconazole (Pfizer SA, Madrid, Spain), posaconazole (Schering-Plough Research Institute, Kenilworth, NJ, USA), and isavuconazole (Basilea Pharmaceutica, Basel, Switzerland). The final concentrations tested ranged from 0.03 to 16 mg/L for amphotericin B and 0.015 to 8 mg/L for the four azoles. *A. flavus* ATCC 204304 and *A. fumigatus* ATCC 204305 were used as quality control strains in all tests performed. Minimal inhibitory concentrations (MICs) were visually read after 24 and 48 h of incubation at 37 °C in a humid atmosphere. MICs were performed at least twice for each isolate. Clinical breakpoints for interpreting AFST results established by EUCAST [22] were used for classifying the *A. fumigatus* strains as susceptible or resistant.

### 2.4. Environmental Surveillance

Our local surveillance program consists of monthly environmental air sampling in operating rooms, ICUs, and high risks units, including the hematology (adult and pediatric) units and the transplantation units (heart, kidney, and liver) for quantitative and qualitative identification of filamentous fungi. Additional samples were also obtained when a suspicious case of *Aspergillus* infection was detected.

The distance between the west walls of UCI 1 and the east walls of UCI 4 is 230 m. All ICUs are located at level +1, with the exception of ICU 9, which was located at level −1.

All the Units in the Critical Areas (2 IPSRU and 9 ICUs) and all the Operating Rooms have independent and double air ducts. All of these double-ducted facilities have been under negative pressure (−15 to −5 Pa) since the start of the SARS-CoV2 pandemic (Figure 1).

During the study period, the attending physicians worked independently in each intensive care unit.

Volumetric air samples from environmental rooms and intensive care units were obtained using a volumetric sampler (Merck Air Sampler MAS100) as previously described [23].

Sealed Sabouraud-dextrose irradiated plates were incubated at 30 °C for 5 days. The plates were examined daily to check for fungal growth. Colonies of *A. fumigatus* growing on the plates were isolated, identified and stored.

### 2.5. Aspergillus fumigatus Strains

In this study, a total of 43 *A. fumigatus* strains were analyzed, 35 clinical and 8 environmental isolates. Strains identification was confirmed by amplification and sequencing of the ITS1-5.8S-ITS2 rDNA regions and a portion of the β-tubulin gene [24].

#### 2.5.1. Cyp51A Amplification, PCR Conditions and Sequencing

For DNA extraction, conidia from each strain were cultured in glucose-yeast extractpeptone (GYEP) liquid medium (0.3% yeast extract, 1% peptone; Difco, Soria Melguizo, Madrid, Spain) with 2% glucose (Sigma-Aldrich Química, Madrid, Spain) for 24 h at 37 °C. After mechanical disruption of the mycelium by vortex-mixing with glass beads, genomic DNA of isolates was extracted using the phenol-chloroform method [25,26]. The full coding sequence of cyp51A including its promoter was amplified and sequenced. To exclude the possibility that any change identified in the sequences was due to PCR-induced errors, each isolate was independently analyzed twice. PCR reaction mixtures contained 0.5 µM of each primer, 0.2 µM of deoxynucleoside triphosphate (Roche, Madrid, Spain), 5 µL of PCR 10× buffer, 2 mM of MgCl_2_, DMSO 5.2%, 2.5 U of Taq DNA polymerase (Applied Biosystems, Foster City, CA, USA), and 100–200 ng of DNA in a final volume of 50 µL. A DNA 1-kb molecular ladder (Promega, Madrid, Spain) was used for all electrophoresis analyses. Samples were amplified in a GeneAmp PCR System 9700 (Applied Biosystems, Foster City, CA, USA). The parameters used were 1 cycle of 5 min at 94 °C and then 35 cycles of 30 s at 94 °C, 45 s at 56 °C for cyp51A promoter and 58 °C for cyp51A gene, and 2 min at 72 °C, followed by a 1 final cycle of 5 min at 72 °C. The amplified products were purified using IllustraExoProStar 1–step (GE Healthcare Life Science, Buckinghamshire, UK), and both strands were sequenced with the Big-Dye terminator cycle sequencing kit (Applied Biosystems, Foster City, CA, USA) following the manufacturer’s instructions. All gene sequences were edited and assembled using the Lasergene software package (DNAStar Inc., Madison, WI, USA). Primers used to amplify and sequence cyp51A and its promoter have been previously described [27].

#### 2.5.2. Strains Genotyping

All of the strains included in this study were genotyped following the previously described typing method TRESPERG [28,29]. Four markers were used: (i) Afu2g05150 encoding an MP-2 antigenic galactomannan protein (MP2); (ii) Afu6g14090 encoding a hypothetical protein with a CFEM domain (CFEM); (iii) Afu3g08990 encoding a cell surface protein A (CSP) and (iv) Afu1g07140 (ERG), which encodes a putative C-24 (28) sterol reductase. The combination of the genotypes obtained with each marker has a discriminatory value (D) of 0.9972 using the Simpson index [30].

## 3. Results

### 3.1. Clinical A. fumigatus Genotypes

The different genotypes isolated from 28 patients during the study period are shown in Table 1. Among the 56 clinical *A. fumigatus* strains isolated from patients, 28 different genotypes were detected. Strains from the same patient hosting the same genotype are summarized only in one representative isolate and similar isolates obtained from the same sample are excluded in the table.

Of the 28 patients analyzed, 22 (78.6%) had only one genotype of *A. fumigatus*. However; the remaining patients harbored more than one genotype of *A. fumigatus*; specifically, five (17.8%) of the patients (14, 17, 22, 24, and 26) harbor in their lungs two different genotypes; and in patient 12 (3.6%) coexisted three different genotypes of *A. fumigatus.* (Table 1).

### 3.2. Hospital Environmental A. fumigatus Genotypes

The chronological distribution of the environmental *A. fumigatus* isolates and level of *A. fumigatus* conidia obtained in the different rooms and ICUs are summarized in Table 2. Out of a total of 336 environmental samples taken during 2020 at the hospital, only seven showed positive growth of *A. fumigatus* (2%) with a minimum count of 1 CFU/m^3^ each one (Figure 1).

During all the study period, only 1 CFU/m^3^ of *A. fumigatus* was isolated in ICU 5 and ICU 6. The remaining ICUs did not show any growth of *Aspergillus*.

Among the seven environmental *A. fumigatus* isolates, seven different genotypes were identified (Table 2).

### 3.3. Correlation between Clinical and Environmental A. fumigatus Genotypes

There was only one coincidence of genotypes between patient and environment during the study period; although there was no correlation in time or location. Specifically, patient 22 (diagnosed on 20 November 2020, on the fifth floor, room 532 B) in whom two different genotypes of *A. fumigatus* coexisted, one of them (t01m1.1g08A.e07) matched with the one isolated in the hospital environment (low level, box 54 on 30 October 2020), being the time difference of 25 days and a location difference of five floors (Table 3).

Regarding matches between patients were as follows:Patient 8 admitted to Pneumology SE 2° B on 8 October 2020, exhibited the same *A. fumigatus* genotype (t03m1.1g05A.e09) as patient 19 admitted to box 36 on 20 November 2020; however, there was a significant difference over time (44 days) and in unit location (2 floors).Patient 9 admitted to box 39 on 8 October 2020, exhibited the same *A. fumigatus* genotype (t02m1.1g09.e16) as patient 14 was admitted to box 47 on 7 November 2020; however, there was a significant difference over time (31 days) and in unit location (different ICUs).Four patients (2, 18, 27 and 28) share the same genotype (t02m1.1g09.e05). Analyzing the patients individually, we observed that there is no correlation neither in time nor in their location in the hospital. The first patient (2) diagnosed on 30 March 2020, in box 10 presents a significant difference of 228 days with the second patient (18) admitted in box 91. The difference with the third patient (27) admitted in box 57 was 254 days, and the difference in time with the fourth patient (28) admitted in box 18 was 279 days. The four patients were admitted to different ICUs.

In summary, although several patients harbored the same *A. fumigatus* genotype, none of them were at the same time and/or in the same unit.

### 3.4. Aspergillus PCR in the Primary Respiratory Sample

*Aspergillus* PCR could be performed on the primary respiratory sample (sputum, tracheal aspirate, and/or bronchoalveolar lavage) obtained after admission, either on the ward and/or in the ICU.

Of the primary samples available from 20 patients, in 18 patients the *Aspergillus* PCR was positive with a CT < 36; and in the other two patients in whom the *Aspergillus* PCR was negative with a cycle greater than 36, the presence of *Aspergillus* was detected with an average of 110 copies/mL, an amount that increased in the following respiratory sample whose *Aspergillus* PCR was already positive.

This data supports the presence and/or co-infection of SARS-CoV-2 and *Aspergillus* since the first day of hospital admission.

### 3.5. Antifungal Susceptibility and Cyp51A Amplification

All isolates were susceptible to all antifungals tested and none of them showed any mutation responsible for azole resistance.

## 4. Discussion

Our study supports the idea that, in our population, COVID-associated aspergillosis is not a nosocomial infection, but a community acquired one, with home environment being the main suspected source of infection.

There are several findings that directly and indirectly back this claim. Firstly, the lack of epidemiological relationship between *A. fumigatus* isolates from hospital environmental air and clinical genotypes reinforce the role of community environmental air in the acquisition of CAPA. In our study, there was only one identical air-patient genotype that, however, did not coincide in time nor in location.

Secondly, during the study period, our hospital environment monitoring has shown excellent air quality, with an *Aspergillus* load of <0.005 CFU/m^3^. Only two ICUs had 1 CFU/m^3^
*Aspergillus* conidia, and they were not related to any of the cases.

It is important to highlight that all the isolates were prospectively collected, which is one of the strengths of our study. We have been able to genotype environmental and clinical isolates, and we have also used a high-discriminatory molecular typing tool (TRESPERG), without finding any matches between environmental hospital air and clinical genotypes. The wide diversity of *A. fumigatus* genotypes among patients and hospital environmental air shows an absence of specific clonal populations in the clinical setting, and thus also supports the hypothesis of a community acquired infection.

There is, as yet, no agreement about how to define a nosocomial case of IA, and different authors have adopted different standards. Patterson et al. [31] defined a nosocomial case of the disease as one that occurred > 1 week after admission to the hospital or <2 weeks after discharge. Although nosocomial outbreaks of IA have contributed to the current perception that most cases of IA are hospital acquired, the short time interval between admission and infection suggests that the patients are colonized with *Aspergillus* before they enter to the hospital, and a significant number of *Aspergillus* infections are acquired on an outpatient basis.

Indeed, the potential impact of *Aspergillus* colonization before hospitalization remains an issue that is still broadly discussed [17,32]; and the role of previous colonization with *Aspergillus* needs to be taken into account while also looking at risk factors related to the host (i.e., immunodepression, underlying diseases, etc.) as well as environmental factors (i.e., airway ventilation, home environment, etc.).

One of the complexities of this diagnosis is that the incubation period for CAPA patients is usually unknown and probably varies among different patients, making it hard to standardize a definition. One of the variables to consider is the fungal load of the previously colonized patient. Logically, with a higher fungal load of *Aspergillus* colonization and a greater viral load of SARS-CoV-2, the progression from colonization to clinical infection could be faster; and, if the patient also had immunosuppression (older, chronic lung disease, etc.) the time elapsed until clinical presentation of invasive aspergillosis could be shorter. Another risk factor to be considered is previous chronic lung damage: in the study by Prattes et al. [5], lung damage was described as a risk factor for the appearance of CAPA, and in the study published by García-Clemente et al. [11], 23% of patients with CAPA had some prior chronic respiratory disease, with these diseases remaining an independent risk factor for CAPA diagnosis in multivariate analysis. It is possible that previous colonization by *Aspergillus* in the lungs of patients with chronic pulmonary diseases could lead to invasive aspergillosis by adding the immune dysregulation and immunosuppression derived from severe SARS-CoV-2 disease. Based on these previous hypothesis, our study shows frequent co-infection of SARS-CoV-2 and *Aspergillus* in the first respiratory sample obtained from a patient on his first hospital admission day, which proves that there are patients previously colonized and/or infected by *Aspegillus*, in their own home and/or outside the hospital, whose symptoms appeared when, due to the severity of the viral infection, they became immunosuppressed.

If we considered how the progression of invasive aspergillosis can be divided into four stages [33,34,35], the majority of CAPA patients in our hospital would be in scenario 2 (Figure 2), namely, community *Aspergillus* colonization, community COVID disease, and hospital diagnosis of community acquired-CAPA. In some patients, the diagnosis of CAPA was even diagnosed post-mortem and there was not time to implement antifungal treatment. This is the main reason why we must advocate for performing a fungal screening to look for an early mycological diagnosis in the primary respiratory sample in all patients with risk factors (older, with chronic lung disease, …) [11] in order to improve the survival of CAPA patients.

Finally, the last indirect evidence that supports our hypothesis that CAPA patients acquire aspergillosis in the community is that, in the first wave of COVID-19 in Spain, all people were housebound and therefore, the main focus of exposure and subsequent colonization would be the patients’ own homes, outside of the hospital setting.

Our study emphasizes the need for preventive measures outside of the hospital and, since the incidence of community *Aspergillus* infections will probably continue rising as high risk patients spend more time outside the hospital setting, we believe that taking care of the home environment of these patients would be essential.

The main limitation of our study is that we have not taken environmental samples at patients’ homes, but we have obtained direct evidence that the IA of our patients was not hospital-acquired since there were no matches between hospital-patient genotypes. Specific host and environmental risk factors still need to be studied in greater detail for CAPA patients.

## 5. Conclusions

To the best of our knowledge, this is the first study monitoring and genotyping *A. fumigatus* isolates from hospital air and COVID-19 patients admitted with aspergillosis, obtained during one year. Our study reveals a wide diversity of *A. fumigatus* genotypes among patients and the absence of specific clonal populations in the clinical setting, and highlights that COVID-associated aspergillosis is not a nosocomial infection; on the other hand, our data support the hypothesis of community acquisition, having home environment (pandemic period at home) as the main suspected focus of infection.

In the future, the role of prior colonization by *Aspergillus* needs prospective studies, since when IA appears during hospitalization, it is difficult to decide whether it is an acquired infection or really an infection that manifests itself during hospitalization starting from an underlying state of previous colonization, that progresses due to the situation of immunosuppression of the admitted patient, and studies must be carried out that take into account the host (immunosuppression, underlying disease, etc.) and environmental situation (ventilation, exposure to water supplies, etc.) characteristics that may be related to the risk of IA.

## Figures and Tables

**Figure 1 jof-08-00316-f001:**
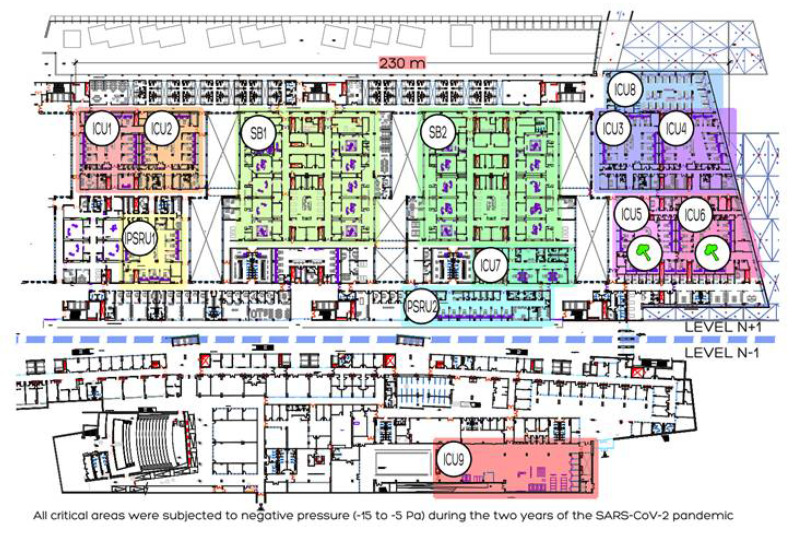
Map of Intensive Care Units (ICU), Intensive Post-Surgical Resuscitation Units (PSRU) and Operating Rooms (SB). Level of *A. fumigatus* conidia load obtained in ICUs 5 and 6 with 1 CFU/m^3^ each one during the study period.

**Figure 2 jof-08-00316-f002:**
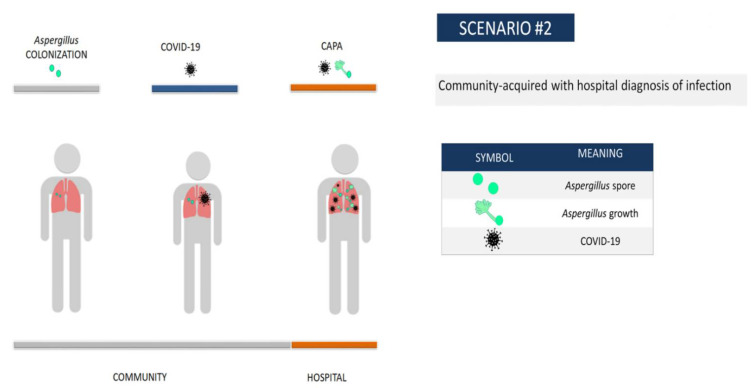
Scenario 2—Community *Aspergillus* colonization, community COVID disease, and hospital diagnosis of community acquired-CAPA.

**Table 1 jof-08-00316-t001:** *A. fumigatus* TRESPERg genotypes of isolates obtained from patient samples and their location at the hospital.

SAMPLE	PATIENT	DATE	LOCATION	GENOTYPE
H-1880	1	23 March 2020	BOX 45	t18bm6.3g09.e09
H-1882	2	30 March 2020	BOX 10	t02m1.1g09.e05
H-1883	3	31 March 2020	BOX 44	t01m1.1g09.e07
H-1885	4	4 April 2020	915 B	t01m5.1g09.e07
H-1891	5	4 April 2020	BOX 29	t09m1.1g08A.e07
H-1918	6	27 April 2020	BOX 54	t28m1.1g09.e20
H-1935	7	16 May 2020	513 A	t03m1.3g08A.e07
H-2096	8	8 October 2020	SE 2° B	t03m1.1g05A.e09
H-2097	9	8 October 2020	BOX 39	t02m1.1g09.e16
H-2104	10	14 October 2020	BOX 54	t03m3.3g05A.e07
H-2129	11	23 October 2020	BOX 32	t04Am1.1g05A.e07
H-2135	12	25 October 2020	BOX 14	t11m1.2g09.e13
H-2136	12	25 October 2020	BOX 14	t03m1.1g10.e06
H-2159	12	3 November 2020	BOX 14	t02m1.8g09.e05
H-2152	13	30 October 2020	BOX 49	t06Bm6.1g08A.e09
H-2158	14	4 November 2020	BOX 47	t06Bm3.4g08A.e11
H-2169	14	7 November 2020	BOX 47	t02m1.1g09.e16
H-2167	15	5 November 2020	BOX 33	t03m1.1g04.e07
H-2175	16	9 November 2020	BOX 24	t04Bm1.2g12.e15
H-2184	17	13 November 2020	BOX 89	t04Am3.4g08A.e11
H-2185	17	13 November 2020	BOX 89	t01m5.1g09.e06
H-2186	18	13 November 2020	BOX 91	t02m1.1g09.e05
H-2194	19	20 November 2020	BOX 36	t03m1.1g05A.e09
H-2203	20	19 November 2020	925 A	t03m1.1g10.e09
H-2211	21	31 November 2020	631 B	t02m14.1g09.e05
H-2217	22	24 November 2020	532 B	t01m1.1g08A.e07
H-2224	22	27 November 2020	306 B	t03m1.1g09.e07
H-2238	23	3 December 2020	BOX 67	t02m1.2g09.e05
H-2239	24	4 December 2020	BOX 26	t04Am3.3g17.eND
H-2267	24	16 December 2020	BOX 31	t04Am3.3g24.eND
H-2244	25	8 December 2020	BOX 84	t01m3.4g05A.e07
H-2245	26	9 December 2020	BOX 40S	t11m13.1g08A.e16
H-2257	26	13 December 2020	BOX 40S	t01m5.1g09.e13
H-2246	27	8 December 2020	BOX 57	t02m1.1g09.e05
H-2279	28	1 January 2021	BOX 18	t02m1.1g09.e05

**Table 2 jof-08-00316-t002:** *A. fumigatus* TRESPERg genotypes of isolates from environmental samples.

SAMPLE	DATE	LOCATION	CFU/m^3^	GENOTYPE
HUCA-1800	23 January 2020	HB 902	1	04Am1.3g08A.e07
HUCA-1801	23 January 2020	HB 902	1	t04Am1.1g04.e07
HUCA-1903	14 April 2020	UCI 6	1	t03m1.1g09.e09
HUCA-2011	29 July 2020	HB 902	1	t02m11.1g09.e16
HUCA-2061	10 September 2020	Cytogenetics	1	t03m1.3g08A.e09
HUCA-2130	19 October 2020	Hb 903	1	t02m1.1g09.e13
HUCA-2151	30 October 2020	UCI 5	1	t01m1.1g08A.e07

**Table 3 jof-08-00316-t003:** *A. fumigatus* isolates with the same TRESPERg genotype.

SAMPLE	ORIGIN	DATE	LOCATION	GENOTYPE
HUCA-2151	Air Sample	30 October 2020	UCI-5 BOX 54	t01m1.1g08A.e07
H-2217	Patient 22	24 November 2020	532 B	t01m1.1g08A.e07
H-2096	Patient 8	8 October 2020	SE 2° B	t03m1.1g05A.e09
H-2194	Patient 19	20 November 2020	BOX 36	t03m1.1g05A.e09
H-2097	Patient 9	8 October 2020	BOX 39	t02m1.1g09.e16
H-2169	Patient 14	7 November 2020	BOX 47	t02m1.1g09.e16
H-1882	Patient 2	30 March 2020	BOX 10	t02m1.1g09.e05
H-2186	Patient 18	13 November 2020	BOX 91	t02m1.1g09.e05
H-2246	Patient 27	8 December 2020	BOX 57	t02m1.1g09.e05
H-2279	Patient 28	1 January 2021	BOX 18	t02m1.1g09.e05

## Data Availability

Not applicable.

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
