# Peer review of "COVID-19 Associated Pulmonary Aspergillosis (CAPA): Hospital or Home Environment as a Source of Life-Threatening *Aspergillus fumigatus* Infection?"

_jof, 2022, doi:10.3390/jof8030316_

Round 1

Reviewer 1 Report

The authors describe the study of the origin of CAPA isolates being hospital acquired or environmental. An interesting and relevant topic to investigate. The study is described in a clear way although a sampling strategy is missing in the methods and the presentation of the data could be improved (see comments).  

Line 25: What do the authors mean with acknowledged? Detected?

Line 57 “is easy to suspect” Do the authors mean that immunosuppressed patients are at risk?

Line 71 “the severity of the COVID-19 clinical picture” do the authors mean that it is associated with a worse outcome/survival? Or length of disease? Clinical picture is very vague.

Line 72 “It has an important impact on the prognosis of the disease” Does a covid infection worsen the CAPA outcome or the other way? Which disease is meant here in this sentence?

Line 179 “2.4 Environmental surveillance” I am missing a description of the strategy of the air samples. Was this done upon admission and the repeated during regular intervals? So what was the timing or plan here? Was it done in the same way for the different patients? Or due to practicalities more on a ad hoc basis? Later in the manuscript it is indicated that 336 environmental samples were taken, that is quite a lot, so hence my question on which strategy was followed here.

Line 210 “2.5.2 Strains genotyping”. The authors state the discriminatory value of the described methods but I wonder why whole genome sequencing was not performed here. The sample size is not so big and I miss a motivation why these markers are appropriate to do such an analysis. Or why not use this as a pre-screening and then whole genome sequence the 10 strains in table 3, were similarities have been found. In this case you can actually compare the two methods and make a conclusion on the usability of these markers as genotyping.

Line 229 and line 237 Tabla -> Table

Line 230 the genotype names does not provide any useful information for the readers, how many SNPs difference do they represent?

Line 238 Figure 1, the figure is very abstract and misses essential data to be of any use for the reader. How many meters are in between the blocks of units? Are they connected via hallways or on different levels? Is there under or overpressure? Does staff/people move in between these units?The symbol for A. fumigatus, how many isolates were found and at what time interval?

Line 249 Table 3 Why not WGS these isolates? And also show whether they are ‘the same’ or not, maybe they have many SNPs differences in other genes that they are actually different.

Line 287 matches -> matched

Author Response

Comments and Suggestions for Authors

The authors describe the study of the origin of CAPA isolates being hospital acquired or environmental. An interesting and relevant topic to investigate. The study is described in a clear way although a sampling strategy is missing in the methods and the presentation of the data could be improved (see comments). 

Line 25: What do the authors mean with acknowledged? Detected? It has been corrected

Line 57 “is easy to suspect” Do the authors mean that immunosuppressed patients are at risk? Yes, it means that immunosuppressed patients are at risk, we think that the sentence is clear

Line 71 “the severity of the COVID-19 clinical picture” do the authors mean that it is associated with a worse outcome/survival? Or length of disease? Clinical picture is very vague. It has been corrected

Line 72 “It has an important impact on the prognosis of the disease” Does a covid infection worsen the CAPA outcome or the other way? Which disease is meant here in this sentence? The sentence has been changed

Line 179 “2.4 Environmental surveillance” I am missing a description of the strategy of the air samples. Was this done upon admission and the repeated during regular intervals? So what was the timing or plan here? Was it done in the same way for the different patients? Or due to practicalities more on a ad hoc basis? Later in the manuscript it is indicated that 336 environmental samples were taken, that is quite a lot, so hence my question on which strategy was followed here.

Our hospital is a reference hospital in the Principality of Asturias, with a large number of operating rooms and high-risk units, all of them monitored at the environmental level in compliance with the standards of  “Validation and evaluation of controlled environment rooms in hospitals” (AENOR-UNE 171340)

Our local surveillance program consists of monthly environmental air sampling in operating rooms, ICUs, and high risks units, including the haematology units for quantitative  and qualitative identification of filamentous fungi. Additional samples were also obtained when a suspicious case of Aspergillus infection was detected in all the patients. This fact explains the large number of environmental samples evaluated.

Line 210 “2.5.2 Strains genotyping”. The authors state the discriminatory value of the described methods but I wonder why whole genome sequencing was not performed here. The sample size is not so big and I miss a motivation why these markers are appropriate to do such an analysis. Or why not use this as a pre-screening and then whole genome sequence the 10 strains in table 3, were similarities have been found. In this case you can actually compare the two methods and make a conclusion on the usability of these markers as genotyping.

R: the reviewer concern is understandable because now a days WGS seems an easy and prompt technology but we have to consider that WGS is quite expensive so that we only use it when is absolutely necessary. Performing a whole genome sequencing analysis for a low number of samples is not an option if the information we are searching could be obtained by other validated methods. That is why we designed the TRESPERG method, which gives a discriminatory power high enough to stablish if two isolates are genetically related. The comparison between these two methods is not a point to be treated in this paper, but some information about the utility of the TRESPERG method can be found in this paper: Garcia-Rubio R, Escribano P, Gomez A, Guinea J, Mellado E. Comparison of Two Highly Discriminatory Typing Methods to Analyze Aspergillus fumigatus Azole Resistance. Front Microbiol. 2018 Jul 20;9:1626. doi: 10.3389/fmicb.2018.01626. At the end, the TRESPERG method was used to determinate whether two isolates are or not the same, information that can be provided with a cheap and fast method like TRESPERG and is not necessary to perform a whole genome sequencing as we don’t need any more information.

Line 229 and line 237 Tabla -> Table

Line 230 the genotype names does not provide any useful information for the readers, how many SNPs difference do they represent?

R: The TRESPERG genotype does not provide any information only with the genotype number. It classifies the genes included in this method attending on their structure, depending on the number and composition of their coding sequence, which has tandem repeats insertions that can be repeated a variable number of times along the structure of the gene. So the differences are not based on SNPs but on tandem repeats and the information can only be understood by comparing the genotypes of the different strains, not by its own. Here we list the two papers that describe the methods and explain how it works, how to interpretate it and tis advantages and limitations:

  • Garcia-Rubio R, Gil H, Monteiro MC, Pelaez T, Mellado E. A New Aspergillus fumigatus Typing Method Based on Hypervariable Tandem Repeats Located within Exons of Surface Protein Coding Genes (TRESP). PLoS One. 2016 Oct 4;11(10):e0163869. doi: 10.1371/journal.pone.0163869.
  • Garcia-Rubio R, Escribano P, Gomez A, Guinea J, Mellado E. Comparison of Two Highly Discriminatory Typing Methods to Analyze Aspergillus fumigatus Azole Resistance. Front Microbiol. 2018 Jul 20;9:1626. doi: 10.3389/fmicb.2018.01626.

Line 238 Figure 1, the figure is very abstract and misses essential data to be of any use for the reader. How many meters are in between the blocks of units? Are they connected via hallways or on different levels? Is there under or overpressure? Does staff/people move in between these units?The symbol for A. fumigatus, how many isolates were found and at what time interval?

The Figure 1 has been changed by a new one realized by the Maintenance Department of the hospital, with a detailed description of all the hospital's areas.

The distance between the west walls of UCI 1 and the east walls of UCI 4 is 230 meters. All ICUs are located at level +1, with the exception of ICU 9, which was located at level -1.

All the Units in the Critical Areas (2 IPSRU and 9 ICUs) and all the Operating Rooms have independent and double air ducts. All of these double-ducted facilities have been under negative pressure (-15 to -5 Pa) since the start of the SARS-CoV2 pandemic. (Fig.-1)

During the study period, the attending physicians worked independently in each intensive care unit.

During all the study period, only 1 CFU/m3 of Aspergillus fumigatus was isolated in ICU 5 and ICU 6. The remaining ICUs did not show any growth of Aspergillus. The chronology and location is described in Table 2 and shown in Figure 1.   

Line 249 Table 3 Why not WGS these isolates? And also show whether they are ‘the same’ or not, maybe they have many SNPs differences in other genes that they are actually different.

R: As I have said before, the WGS is really expensive for such a small number of isolates and it does not provide any information that can not be obtained with the TRESPERG method, so this method, faster and cheaper, is enough for our intentions. Reviewer is right, maybe there are several SNPs between strains, that can emerge only by the mistakes that the polymerase makes in every mitosis, but it does not indicate that they are different strains. However, our method has proven to give a discriminatory index enough to differentiate between isolates only with those four genes, and this is due to the special structure of these genes.

Line 287 matches -> matched

Reviewer 2 Report

The authors aimed to present an analysis regarding COVID-19 associated Pulmonary Aspergillosis, focusing particularly on the origin of the fungal infection. Considering the severity of the pandemic situation, this article is offering  some interesting observation, is fully up to date, and I am surprised about the low number of publications focusing on similar aspects. Hence, I deeply appreciate the authors' idea and effort in such project.

I would have only few minor remarks and, some more concern regarding the style of the discussion part.

The introduction is clear, well written and easy to follow, as the materials and methods and results.

I am not fully sure I understand the role played in this study from Cyp51A and its sequencing (before even performing the sequencing, a simple MIC test would have revealed no azole resistance and hence the sequencing of Cyp51A and its promoter would have been useless...). Moreover, why using a Taq polymerase for products that should undergo sequencing instead of a proofreading polymerase?

The discussion part shows big differences in style and, particularly, a lower lever of English (several grammar mistakes, third person singular with wrong verb, pointed list - first second third - not presented in the correct way. It is possible to understand the message through the text, but it seems a constant reiteration of the same big point, in a very confusing style and organization. (It almost looks as the discussion was written by a completely different person).

I would hope the authors could offer a little proofreading/reorganization job on this particular part; after that I would fully support the publication of their work.

Author Response

Reviewer 2

Comments and Suggestions for Authors

The authors aimed to present an analysis regarding COVID-19 associated Pulmonary Aspergillosis, focusing particularly on the origin of the fungal infection. Considering the severity of the pandemic situation, this article is offering some interesting observation, is fully up to date, and I am surprised about the low number of publications focusing on similar aspects. Hence, I deeply appreciate the authors' idea and effort in such project.

I would have only few minor remarks and, some more concern regarding the style of the discussion part.

The introduction is clear, well written and easy to follow, as the materials and methods and results.

I am not fully sure I understand the role played in this study from Cyp51A and its sequencing (before even performing the sequencing, a simple MIC test would have revealed no azole resistance and hence the sequencing of Cyp51A and its promoter would have been useless...). Moreover, why using a Taq polymerase for products that should undergo sequencing instead of a proofreading polymerase?

R: We use a Taq polymerase to amplify the gene and a proofreading for sequencing it, to date we can not amplify and sequence at a time so here the Taq polymerase is used only for amplification. In any case, we normally repeat gene amplification and sequencing when we found any mutation before to conclude that any mutation found is responsible for a resistant phenotype.

The explanation of why we do Cyp51a sequencing is that we do the search for any mutations in Cyp51A looking for new mutations (even if they are polymorphisms no responsible for resistance phenotypes) or non-synonymous mutations that could explain the acquisition of initial mutations or TR integrations in combined resistance mechanisms such as (TR34/L98H or TR46/Y121F/A289T) in the case the acquisition of this combined mutations is selected in a two-step process. However, the reviewer is right in the sense that in this work the study of Cyp51A did confirm the absence of mutations, and therefore could be excluded in this manuscript.

The discussion part shows big differences in style and, particularly, a lower lever of English (several grammar mistakes, third person singular with wrong verb, pointed list - first second third - not presented in the correct way. It is possible to understand the message through the text, but it seems a constant reiteration of the same big point, in a very confusing style and organization. (It almost looks as the discussion was written by a completely different person).

I would hope the authors could offer a little proofreading/reorganization job on this particular part; after that I would fully support the publication of their work.

The discussion has been modified. We hope that the new discussion will be to the reviewer's approval in terms of style, organization and level of English.